# MicroRNAs in Epithelial–Mesenchymal Transition Process of Cancer: Potential Targets for Chemotherapy

**DOI:** 10.3390/ijms22147526

**Published:** 2021-07-14

**Authors:** Fu Peng, Huali Fan, Sui Li, Cheng Peng, Xiaoqi Pan

**Affiliations:** 1Department of Pharmacology, Key Laboratory of Drug-Targeting and Drug Delivery System of the Education Ministry, Sichuan Engineering Laboratory for Plant-Sourced Drug and Sichuan Research Center for Drug Precision Industrial Technology, West China School of Pharmacy, Sichuan University, Chengdu 610041, China; fujing126@yeah.net (F.P.); hwari_f@163.com (H.F.); lisui98@163.com (S.L.); 2State Key Laboratory of Southwestern Chinese Medicine Resources, Chengdu University of Traditional Chinese Medicine, Chengdu 611137, China

**Keywords:** miRNA, cancer, epithelial–mesenchymal transition, drug candidates

## Abstract

In the last decades, a kind of small non-coding RNA molecules, called as microRNAs, has been applied as negative regulators in various types of cancer treatment through down-regulation of their targets. More recent studies exert that microRNAs play a critical role in the EMT process of cancer, promoting or inhibiting EMT progression. Interestingly, accumulating evidence suggests that pure compounds from natural plants could modulate deregulated microRNAs to inhibit EMT, resulting in the inhibition of cancer development. This small essay is on the purpose of demonstrating the significance and function of microRNAs in the EMT process as oncogenes and tumor suppressor genes according to studies mainly conducted in the last four years, providing evidence of efficient target therapy. The review also summarizes the drug candidates with the ability to restrain EMT in cancer through microRNA regulation.

## 1. Introduction

Epithelial–mesenchymal transition (EMT) refers to a process by which epithelial cells change into mesenchymal stem cells with an increase in migratory and invasive capacities [1]. EMT has been categorized into three different subtypes based on the distinguished biological settings as well as functional consequences. Among them, the “type 3” is associated with cancer development, also called as oncogenic EMT (Figure 1). Epithelial-like cells are non-mobile with adherent junctions, while mesenchymal-like cells, devoid of cell polarization, could promote the invasion and migration of primary tumor cells from one site to a distant site [2]. E-cadherin is considered as the epithelial marker, modulating cell–cell adhesion [3]. N-cadherin and vimentin are identified as mesenchymal markers. E-cadherin to N-cadherin switch in cancer leads to a more aggressive migration, invasion, and metastasis [4]. Snail and Slug transcription factors could promote epithelial cells migrating and translating into mesenchymal cells via inducing an E-cadherin to N-cadherin switch [5,6]. The mesenchymal-to-epithelial transition (MET) is the opposite process of EMT, namely, converting mesenchymal cells to epithelial cells [7]. MET allows cancer cells to colonize and form a potentially deadly metastatic lesion at the secondary tumor site [8]. EMT transcription factors (EMT-TFs) coordinate the EMT process. Among them, EMT inducers (*SNAI1/2*, *ZEB1/2*, *TWIST1/2*, etc.) are the most widely studied, with the capacity to suppress E-cadherin and promote mesenchymal transition [9]. In addition, some transcription factors identified as EMT suppressors (*GRHL2*, *OVOL1/2*, etc.) could drive the MET process [10]. According to the bioinformation analysis, *GRHL2* is associated with epithelial characteristics across various cancers, inducing mesenchymal phenotypes [11]. Meanwhile, OVOL proteins function as the suppressors of EMT-inducing transcription factors and deletion of *OVOL1/2* will result in EMT progression [12].

MicroRNAs, also called miRNAs, are a kind of small non-coding RNA molecules of about 22 nucleotides which function as negative regulators to inhibit the expression of target genes [13]. With the base-pairing capability, miRNAs could majorly bind to the 3′ untranslated region (UTR) of its target genes, resulting in the degradation of target mRNAs or translational inhibition. Specifically, the perfect complementarity of miRNA and its targets tends to result in degradation of the targeted mRNA, while the imperfect complementary binding indicates the translational inhibition of the targets [14,15]. One miRNA could even interact with numerous mRNAs, and multiple miRNAs could regulate target genes cooperatively, which enables miRNAs to be the most substantial regulators in cells [16]. Generally speaking, miRNAs in cancers could affect cell proliferation, migration, invasion, stemness, drug resistance, angiogenesis, tumor growth, even metastasis, etc. [17]. Since a great number of studies gradually started to identify functions of miRNAs on the EMT process of cancer, miRNAs were found to play critical roles as both oncogenic genes (oncomiRs) and tumor suppressive genes (TsmiRs) in the development of EMT [18,19,20]. 

In the last decade, the cancer treatment efforts in the field of miRNAs have increased continuously. Although miRNA-based cancer diagnostics and prognostics have been gradually applied in cancer management, miRNA-based therapeutics are still considered as emerging challenges. The main approaches focus on inactivating or abating oncomiRs, activating or promoting tsmiRs, and modulating deregulated miRNAs [21]. Besides single strand interfering RNAs, such as miR MRX34 in a phase I study (NCT01829971), natural compounds have gained attention due to their possible applications in the regulation of miRNAs. Interestingly, these related findings even point out that single compounds are promising drug candidates in that they can regulate multiple miRNAs at the same time [22]. Thus, this review is aimed at illustrating the function and importance of miRNAs in the EMT process as promising targets in cancer treatment and the collection of compounds with the capacity to suppress EMT via regulating microRNA in cancer.

## 2. OncomiRs in EMT

In most conditions, it was found that oncomiRs were overexpressed or upregulated, promoting the development of EMT and giving rise to the metastatic potential. OncomiRs could promote the EMT process through activating *TGFβ*, *FGF*, and Notch signaling pathways, suppressing EMT inhibitors, and enhancing invasion and motility [23].

### 2.1. Breast Cancer

Breast cancer is ranked as the most common cause of cancer-death among women worldwide. The expression of miR-27a was higher in breast cancer tissues and cell lines than the normal counterparts. Overexpressed miR-27a could induce the EMT process and promote migration. *FBXW7* was expressed at a low level in breast cancer tissues and cell lines and rescuing *FBXW7* would at least partly reverse the oncogenic effect of miR-27a [24]. MiR-122-5p was highly expressed in triple-negative breast cancer (TNBC) while *CHMP3* expression was low, which was associated with poor survival in patients with TNBC. At the cellular level, miR-122-5p promoted TNBC cell viability, proliferation, and invasion but inhibited apoptosis by targeting *CHMP3* mRNA. Furthermore, miR-122-5p compelled the EMT process and MAPK signaling in TNBC [25]. MiR-155 was expressed at a high level in breast cancer tissues compared with paired normal tissues, while *TGFBR2* was expressed at a low level. High expression of miR-155 could promote tumor growth and metastasis. Transfecting miR-155 inhibitors could inhibit cell proliferation and even EMT process, and increase *TGFBR2* expression. Luciferase assay exerted that *TGFBR2* was the direct target of miR-155, suggesting that miR-155 might promote breast cancer progression through the down-regulation of *TGFBR2* [26]. High expression of miR-615-3p was determined in invasive breast cancer cells and metastatic breast cancer tissues. Overexpressed miR-615-3p could critically promote cell motility in vitro and pulmonary metastasis in vivo. Interestingly, the study also exerted that *PICK1* was the direct target of miR-615-3p, and *PICK1* inhibited the processing of pre-miR-615-3p to mature miR-615-3p, resulting in a negative feedback loop [27].

### 2.2. Colorectal Cancer

In recent years, the occurrence of colorectal cancer has been gradually increasing with poor overall survival rate [28]. The miR-410-3p expression levels were significantly up-regulated in colorectal cancer (CRC) cells and tissues, while its target, namely, *ZCCHC10* was markedly down-regulated. In the functional assay, miR-410-3p could promote EMT with the enhanced ability of migration and invasion of HT29 and SW480 cells. Interestingly, the increase in *ZCCHC10* could reverse the effect of miR-410-3p [29]. MiR-425-5p promoted CRC tumor growth and metastasis in vivo. Transfection of miR-425-5p mimic in SW480 cells exerted a promotive effect on cell viability, cell cycle entry, migration, invasion, and EMT process. Moreover, high expression of miR-425-5p promoted β-catenin translocating to the nucleus. Transfection of miR-425-5p inhibitor in LOVO cells demonstrated opposite effects. Luciferase reporter assay confirmed that *CTNND1* was the direct target of miR-425-5p. Silencing of *CTNND1* would mimic the effect of miR-425-5p overexpression [30]. RASSF6 hypermethylation has been shown to be frequent in a variety of solid cancers, according to extensive research. A recent study showed that miR-496 was aberrantly up-regulated in CRC and *RASSF6* was a direct target of miR-496. Further research revealed that the stimulation of Wnt signaling by miR-496 targeting *RASSF6* might have facilitated cell migration and EMT, yet had little effect on cell survival. In summary, it was confirmed that miR-496/*RASSF6* axis was engaged in Wnt pathway-mediated tumor metastasis which may be a therapeutic target for CRC [31].

### 2.3. Cervical Cancer

Cervical cancer is one of the common types of cancer occurring among women in developing countries. MiR-150-5p has been found to up-regulate in cervical carcinoma. The research suggested that the levels of miR-150-5p were obviously different in C-33A and HeLa cells. MiR-150-5p was strongly associated with cell growth. Furthermore, *SRCIN1* acted as a direct target of miR-150-5p. When transducing miR-150-5p mimics, the expression of SRCIN1 increased, the proliferation and EMT of cervical cancer cells were suppressed significantly, and apoptosis was accelerated by *SRCIN1* [32]. Recently, a study revealed that the expression of miR-199a-5p was high in cervical carcinoma tissues and cell lines, and thus promoted cell proliferation and metastasis. Meanwhile, the vimentin and N-cadherin expressions were enhanced while E-cadherin was reduced by high levels of miR-199a-5p, which were the significant markers of EMT. Further research indicated that *PIAS3* was the direct targets of miR-199a-5p and was negatively correlated with miR-199a-5p. In contrast, the effects of miR-199a-5p were reversed by up-regulation of *PIAS3* in cervical carcinoma [33].

### 2.4. Gastric Cancer

Evidence showed that the aberrantly high expression of miR-21 contributed to early metastasis in gastric cancer. In this study, miR-21 was higher in gastric cancer cell lines compared with a gastric mucosal epithelial cell line. Furthermore, miR-21 mimic could increase N-cadherin and vimentin, and decrease E-cadherin to promote EMT in gastric cancer cells whereas miR-21 inhibitor showed the opposite effects. As a result, up-regulation of miR-21 facilitated cell metastasis and invasion of MGC-803 cells. The decreased expression of miR-21 suppressed cell metastasis and invasion in gastric cancer [34]. Overexpressed miR-130b-3p was detected in gastric cancer tissues, and miR-130b-3p promoted proliferation, migration, and EMT of gastric cancer cells as well as tumor formation and metastasis in vivo. Additionally, miR-130b-3p could directly target *MLL3*, and *MLL3* could promote the expression and function of *GRHL2*. Decreasing miR-130b-3p or upregulating of *GRHL2* inhibited tumor growth and angiogenesis in gastric cancer [35]. The overall survival rates of advanced gastric cancer are still poor despite the progression of pharmacotherapy [36]. MiR-616-3p was expressed at a high level in cancer tissues, and high expression of miR-616-3p was positively correlated with poor prognosis. Functional assay exerted that overexpressed miR-616-3p enhanced angiogenesis and EMT in gastric cancer cells through the down-regulation of *PTEN*. Furthermore, rescuing *PTEN* could at least partly inhibit the effect of miR-616-3p on gastric cancer cells [37].

### 2.5. Nasopharyngeal Carcinoma

Nasopharyngeal carcinoma is a kind of cancer developed from nasopharyngeal epithelium [38]. A study showed that increased expression of miRNA-10b promoted the growth of nasopharyngeal carcinoma cells CNE1 in vitro by regulating EMT, proliferation, and migration [39]. The study demonstrated that miR-205-5p was significantly higher in the cisplatin-resistant nasopharyngeal carcinoma cell line HNE1/DDP than in its parental cell HNE1. According to the findings, low expression of miR-205-5p inhibited EMT development in HNE1/DDP cells. Further research suggested that down-regulation of the expression of *PTEN*, a downstream candidate gene of miR-205-5p, could counterbalance the impact of miR-205-5p inhibitors. Therefore, miR-205-5p was shown to regulate EMT through the PI3K/AKT pathway by targeting *PTEN* on HNE1/DDP cells [40]. 

### 2.6. Lung Cancer

Lung cancer is considered as one of the most common cancers with extremely poor prognosis over the years. MiR-21 was overexpressed in human non-small cell lung cancer (NSCLC) cells, and overexpressed miRNA-21 induced cyclin D1 and cyclin E1 expression and promoted proliferation of cancer cells. In addition, high expression of miRNA-21 increased the EMT process through *PTEN*/Akt/*GSK**3**β* signaling [41]. MiR-942 was detected expressed at a high level in human NSCLC tissues and cells. Overexpressed miR-942 could promote cell migration, invasion, and angiogenesis. Furthermore, miR-942 was attributed to NSCLC metastasis in tail vein xenograft models. Three bioinformatics software predicted *BARX2* as a downstream target of miR-942, and luciferase assay validated that miR-942 could directly bind to the 3′UTR region of *BARX2*. Western blot results exerted that miR-942 increased EMT-associated proteins by down-regulation of *BARX2* [42]. 

### 2.7. Osteosarcoma

Osteosarcoma is the most common malignant bone tumor occurring in youth [43]. In osteosarcoma tissues and cell lines, there was obviously increasing expression of miR-17-5p, which played a role in suppressing cell proliferation, EMT, and accelerating apoptosis in osteosarcoma. Further study revealed that *SRCIN1*, an anti-oncogene, was served as a target of miR-17-5p. When silencing miR-17-5p, the EMT-related proteins expression changed, and thus regulated EMT progression of osteosarcoma [44].

### 2.8. Other Types of Cancer

Intrahepatic cholangiocarcinoma is ranked as the second most common malignant hepatic tumor with a relatively high postoperative recurrence rate [45]. High expression of miR-19b-3p was detected in intrahepatic cholangiocarcinoma tissues compared with adjacent tissues, and overexpressed miR-19b-3p could promote cell proliferation and EMT, and inhibit apoptosis. Further study exerted that *CCDC6* was a predicted target of miR-19b-3p, and in vivo study displayed that miR-19b-3p/*CCDC6* axis modulated EMT to promote intrahepatic cholangiocarcinoma development [46]. Ovarian cancer is related to gynecological tumors with an extremely poor prognosis [47]. MiR-27a was expressed at a high level in ovarian cancer tissues and cells. Functional assays exerted that overexpressed miR-27a promoted migration and invasion of HO8910 and OV90 cells, while decreasing miR-27a displayed the opposite effect. Furthermore, high expression of miR-27a induced EMT markers and processes. *FOXO1* was found as the direct target of miR-27a, and miR-27a could significantly inhibit *FOXO1* expression as well as the Wnt pathway [48]. Prostate cancer is one of the most common cancers among men, resulting in over 11,000 deaths per year [49]. Real-time PCR results determined the overexpression of miR-19a in prostate cancer. Clinical data suggested a negative correlation between miR-19a expression and poor outcome of patients. Luciferase reporter assay exerted that miR-19a could directly target the mRNA 3′-UTR of *CUL5* in PC3 cells. In addition, decreased *CUL5* expression would predict a worse outcome for prostate cancer patients. Functional assay found that miR-19a promoted cell migration, invasion, EMT, and metastasis in prostate cancer through down-regulation of *CUL5*, while rescuing *CUL5* would partially reverse the roles of miR-19a on prostate cancer [50]. Renal cancer belongs to the most common type of kidney carcinoma, accounting for about 3% of tumors in adults [51]. MiR-155-5p was expressed at a high level in renal cancer, and overexpressed miR-155-5p could promote cell proliferation, colony formation, migration, and invasiveness of renal cancer cells. In contrast, decreasing miR-155-5p could inhibit cancer development. Meanwhile, miR-155-5p could increase N-cadherin and Snail, and promote the EMT process [52].

## 3. TsmiRs in EMT

TsmiRs, as tumor suppressors, show the opposite function of oncomiRs in EMT pathways. TsmiRs were found to be dramatically downregulated or completely eliminated in cancer cells, and the rescue of tsmiRs could inhibit EMT development, with the increase in E-cadherin and decrease in N-cadherin. Some tsmiRs could directly target SNAI, Twist, and ZEB families, inhibiting their expression (Figure 2).

### 3.1. Breast Cancer

Bioinformatics analysis indicated that Gli1 was the direct target of miR-34b. Further study determined that miR-34b was markedly decreased in MCF-7 and MDA-MB-231 cells than in normal mammary gland epithelial MCF-10A cells, while the expression of Gli1 was the opposite. Reducing expression of miR-34b would enhance Gli1, promote the EMT process as well as cell invasion, while increasing its expression could reverse the effects [53]. Real-time PCR results determined that miR-124 was expressed at a low level in TNBC tissues and cells. Overexpressed miR-124 could inhibit proliferation, migration, invasion, and the EMT process of TNBC cells. *ZEB2* was a direct target of miR-124 according to the results of luciferase reporter assay. Rescue assays manifested that miR-124 inhibited EMT and metastasis through the inhibition of *ZEB2* [54]. In vitro and vivo experiments revealed that the ectopic expression of miR-135 obviously suppressed cell proliferation, migration, invasion, and EMT in breast cancer cell lines MDA-MB-468 and MCF-7 cells, and reduced tumor volume. Moreover, mechanistic studies suggested that the miR-135 mimic could down-regulate the N-cadherin and vimentin at mRNA and protein levels, and enhance the expression of E-cadherin. Meanwhile, miR-135 could increase p-GSK3β expression and decrease Wnt and β-catenin expression which suggested that miR-135 partly inactivated the Wnt/β-catenin pathway to suppress breast cancer [55]. It has been observed that *DUSP4* expression was significantly up-regulated in breast cancer cell lines, and the miR-137 level was dramatically down-regulated. Moreover, the introduction of miR-137 significantly inhibited resistance to doxorubicin of breast cancer cells by decreasing EMT. Besides, up-regulation of *DUSP4* in breast cancer cells partially abolished the chemo-resistance of miR-137 mimic. Moreover, the results of miR-137 enhanced the sensitivity of breast cancer cells to doxorubicin in vivo and were consistent with that in vitro [56]. The low expression of miR-186 in breast cancer tissues and cells is related to tumor metastasis and a poor overall survival in breast cancer patients. MiR-186 with high expression suppressed breast cancer cells proliferation, migration, and EMT progression, while miR-186 inhibition showed the opposite results. Moreover, it was confirmed that miR-186 directly targeted Twist1 in breast cancer cells. The activities of miR-186 were attenuated by restoring Twist1 [57]. In this study, researchers found up-regulation of miR-214 inhibited cell proliferation and invasion, while inhibition of miR-214 expression could have the opposite result. In addition, the survival rate of breast cancer patients was favorably connected with miR-214 levels and negatively connected with *RNF8* levels which is a recently discovered regulator that promotes EMT. Further study firstly suggested that miR-214 could inhibit *RNF8* to regulate EMT and thus suppress breast cancer [58]. MiR-516a-3p expression was expressed at a low level in human breast cancer tissues and cells, while the expression of Pygo2 was high. Additionally, the expression of miR-516a-3p was negatively associated with poor prognosis for patients with breast cancer, while the expression of Pygo2 exerted a positive correlation to poor prognosis. Luciferase reporter assay validated Pygo2 as the direct target of miR-516a-3p. Functional study exerted that increasing miR-516a-3p expression could suppress cell growth, migration, invasion, and EMT process, and reducing miR-516a-3p had a reverse effect [59]. MiR-574-5p was down-regulated in breast cancer tissues and cells. MiR-574-5p inhibited proliferation, migration, and EMT in TNBC cells. Furthermore, miR-574-5p had an inhibitory effect on tumor size and metastasis in vivo. Luciferase assay exerted that *BCL11A* and *SOX2* were the direct target of miR-574-5p. The suppressive effect of miR-574-5p in TNBC cells was at least partly due to *SOX2* and *BCL11A* [60]. The up-regulation of miR-6838-5p in TNBC could inhibit cell migration, invasion, and Wnt signaling pathway, while down-regulating miR-6838-5p had a reverse effect. Luciferase reporter and RNA immunoprecipitation assay confirmed that *WNT3A* was the direct target of miR-6838-5p. Interestingly, rescuing *WNT3A* would at least partly reverse the suppressive effect of miR-6838-5p on the TNBC EMT process [61].

### 3.2. Colorectal Cancer

Bioinformatic analysis displayed that miR-1-3p was expressed at a low level in CRC tissues, and real-time PCR results validated the expression in CRC cells. Overexpressed miR-1-3p could inhibit CRC cell proliferation and invasion. Advanced study exerted that miR-1-3p could directly bind to *YWHAZ*, resulting in inhibition of the EMT process [62]. Research suggested that the expression of miR-9-5p was decreased and the expression of *FOXP2* was increased in CRC tissues and cell lines. Moreover, *FOXP2* had oncogenic roles in CRC and miR-9-5p could inhibit cell metastasis and EMT in CRC by targeting *FOXP2*. The low miR-9-5p levels and the high *FOXP2* levels were associated with poor prognosis of CRC patients. Therefore, miR-9-5p/*FOXP2* axis is key to diagnose and treat CRC [63]. Real-time PCR showed that the expression of miR-205 was negatively correlated with *MDM4* in CRC tissues and cells. Luciferase assay displayed a direct binding between miR-205 and *MDM4*. Advanced study demonstrated that overexpressed miR-25 inhibited proliferation, migration, and invasion of HCT116 cells. Additionally, the decreased N-cadherin, vimentin, MMP2, and MMP9 with the increase of E-cadherin were observed in miR-205-overexpressed cells [64]. MiR-145-5p were down-regulated in CRC samples and cell lines. Elevated miR-145-5p expression inhibited cell viability, migration, invasion and EMT of LoVo and SW480 cells, while reducing miR-145-5p displayed the opposite effect. *CDCA3* was confirmed as a direct target of miR-145-5p according to the luciferase reporter assay. In addition, rescuing *CDCA3* in SW480 cells could at least partly reverse the influence induced by miR-145-5p [65]. It was interesting to find that the expression of miR-330 was negatively correlated with HMGA2. The high expression of *HMGA2* and low expression of miR-330 were detected in CRCs with poor long-term patient survival. Evaluated miR-330 expression in HCT116 and SW480 cells suppressed the oncogenic effect of *HMGA2*. Further study exerted that miR-330 could inhibit cell migration and viability, the Akt pathway, as well as the essential regulators in the EMT process through the down-regulation of *HMGA2* [66]. MiR-370-3p played a key role in repressing CRC proliferation and EMT in vitro. The underlying mechanism proved that miR-370-3p could reduce tumor-related inflammatory factors, such as TNF-α, IL-1 β, and IL-6, thereby inhibiting proteins such as P53, β-catenin, and ki67, which are closely related to tumor growth [67]. MiR-873-5p overexpression dramatically inhibited CRC cell proliferation and EMT. Mechanistically, miR-873-5p directly targeted *JMJD8* which was significantly up-regulated in CRC tissues and cell lines to impact the proliferation and EMT process. Finally, the research also found that miR-873-5p could suppress the NF-кB pathway in CRC. Taken together, miR-873-5p could inhibit proliferation and EMT in CRC by targeting *JMJD8* and suppressing the NF-кB signaling pathway, which may provide a new antitumor strategy to carcinogenesis [68]. MiR-3622a-3p was remarkably down-regulated in CRC according to the TCGA database. Functional assays exerted that miR-3622a-3p could inhibit proliferation, apoptosis, cell cycle, migration, and invasion of CRC cells. The results from dual luciferase assay, RNA immunoprecipitation assay, and pull-down assay showed that *SALL4* was the direct target of miR-3622a-3p. Further studies showed that miR-3622a-3p inhibited stemness and the EMT process of CRC cells through *SALL4* mRNA decrease. Additionally, miR-3622a-3p also manifested the anti-cancer effect in a tumor xenograft model and an in vivo metastasis model [69].

### 3.3. Gastric Cancer

Research indicated that the level of miR-105 was low in gastric cancer tissues and cells, and *SOX9* was expressed highly in gastric cancer cells. Further study suggested that miR-105 was negatively correlated with *SOX9*, and directly targeted it to suppress cell migration, invasion, and EMT in gastric cancer. The partial effects of miR-105 on cell migration and invasion could be reversed by *SOX9*. *SOX9* knockdown could inhibit gastric cancer cell migration, invasion, and EMT [70]. In this study, miR-129-5p was expressed lower in gastric cancer cells than in GES-1. In addition, the high levels of miR-129-5p could suppress EMT and proliferation through directly targeting *HMGB1* which was up-regulated in gastric cancer cells [71]. Low expression of miR-203 was detected in gastric cancer tissues and cell lines. Overexpressed miR-203 could suppress invasion and the EMT process. Bioinformatics analysis indicated annexin A4 as the direct target of miR-203, and the binding was validated by luciferase reporter assay. Additionally, rescuing annexin A4 in GC cells at least partly reversed the inhibitory effect of miR-203 [72]. A recent study indicated that the expression of *FOXQ1* was so high in gastric cancer cells that it greatly promoted proliferation, migration, invasion, and EMT. Further study suggested that *FOXQ1* was directly regulated by miR-519 which was observed weakly expressed in both gastric cancer tissues and cells. In other words, miRNA-519 functioned as a suppressor in regulating gastric cancer EMT by targeting *FOXQ1* [73]. Low expression of miR-630 was detected in gastric cancer cell lines. MiR-630 mimic transfection could inhibit migration and invasion in SGC-7901 and BGC-823 cells. Furthermore, EMT phenotype was suppressed after the increase in miR-630 [74]. The decrease in miRNA-665 expression is associated with gastric cancer occurrence. In vitro, miRNA endogenous mimics could repress gastric cancer cell lines growth by targeting the 3′-UTR of the *CRIM1* gene which was found to promote the EMT process in GC cell lines. Interestingly, up-regulating the expression of *CRIM1* would eliminate the inhibitory effect of miRNA-665 on gastric cancer [75]. 

### 3.4. Glioma

Glioblastoma multiforme (GBM) is considered as the most common malignant brain tumor, identified as a Grade IV astrocytoma [76]. Functional assay displayed that high expression of miR-96 could inhibit migration, invasion, and proliferation in GBM cells. Luciferase assay indicated *AEG-1* as the direct target of miR-96. Further study exerted that miR-96 could regulate EMT markers and suppress EMT process through the downregulation of *AEG-1* in GBM cells. Additionally, the down-regulation of *AEG-1* would result in a similar effect as overexpressed miR-96 in GBM [77]. MiR-205 was found to express lowly in glioma tissues and human glioma cell lines U87 and U251. In vitro, the overexpression of miR-205 increased E-cadherin, decreased mesenchymal markers, and inhibited cell proliferation, migration, and invasion. Furthermore, miR-205 was reported to directly bind to *HOXD9* and decreased its expression so that EMT progression was reversed [78]. When glioma tissues from patients were compared with normal cells, the expression of miR-218-5p was shown to be lower. After transfecting miR-218-5p into U251 and U87 cells, the findings showed that miR-218-5p could suppress cell activity, proliferation, and invasive ability as well as EMT. Further research revealed *LHFPL3* was highly expressed in glioma and a direct target of miR-218-5p. Namely, miR-218-5p could bind to *LHFPL3* to inhibit the invasion of glioma cells [79]. The miR-802 level was dramatically down-regulated in GBM tissues, and the *SIX4* expression was significantly up-regulated. Moreover, high *SIX4* expression was strongly associated with a low miR-802 level in GBM tissues. In addition, the introduction of miR-802 significantly inhibited proliferation, invasion, and EMT of GBM cells. Furthermore, knockdown of *SIX4* had similar effects with miR-802 overexpression on GBM cells [80]. In vitro, the expression of miR-876-5p was observed to decrease in all of the GBM cell lines while *TWIST1* expression levels showed the opposite trend. This finding indicated that miR-876-5p inhibited EMT in GBM accompanied with downregulation of gene *TWIST1*, and thus is a promising target for the treatment of GBM [81].

### 3.5. Lung Cancer

The down-regulation of miR-138-5p was detected in lung adenocarcinoma tissues and cell lines. The high levels of miR-138-5p inhibited EMT, proliferation, and metastasis in A549 and H1299 cells. Additionally, miR-138-5p was identified to bind directly to *ZEB2*, and decreased *ZEB2* to suppress the EMT process, proliferation, and metastasis of lung adenocarcinoma cells [82]. In this study, it was verified that miR-145 was associated well with invasive ability as a result of low expression in NSCLC cell lines H1299, PC7, and SPCA-1 cells. When compared with adjacent normal tissue, the expression of miR-145 was lower in cancer tissue. The important markers of EMT, namely, N-cadherin, vimentin, and E-cadherin were down-regulated by overexpression of miR-145. Further research indicated that the oncogene *ZEB2* was bound by miR-145, and miR-145 could decrease *ZEB2* to inhibit EMT progression in NSCLC as well [83]. MiR-146b expression was down-regulated in cisplatin resistant human lung adenocarcinoma cells (A549/DDP and H1299/DDP), which indicated that miR-146b could function as a tumor suppressor in cisplatin resistant human lung adenocarcinoma cells. Furthermore, *PTP1B* was a target of miR-146b. Then, miR-146b suppressed the EMT of H1299/DDP and A549/DDP cells and improved cisplatin sensitivity. Additionally, to detect the impact of miR-146b up-regulation on tumor growth in vivo, xenograft model mice were used, and these findings were consistent with that of in vitro [84]. MiR-335-5p exerted a significant effect on repressing TGF-β1-mediated NSCLC migration and invasion. More importantly, over-expressed miR-335-5p could specifically bind to *ROCK1*, thus inhibiting the EMT, migration, and invasion in NSCLC. Consistently, deceased *ROCK1* efficiently impaired TGF-β1-mediated EMT and migratory and invasive capabilities of NSCLC cells. In other words, overexpression of miR-335-5p inhibited cell proliferation and EMT of NSCLC cells by directly down-regulating *ROCK1* expression [85]. In a recent study, findings showed that the overexpression of miR-363-3p could inhibit EMT to suppress cell migration and invasion in NSCLC while knocking down miR-363-3p had the opposite results. Moreover, it was identified that miR-363-3p bound to 3′UTR of *NEDD9* and *SOX4*, negatively regulating their expression. Intriguingly, *NEDD9* or *SOX4* knockdown could reverse the metastasis-promoting function of antagomiR-363-3p and the inhibitory effects of agomiR-363-3p. Collectively, miR-363-3p may act as an anti-oncogene to be a potential therapeutic target for NSCLC [86]. The lower expression of miR-379 expression in NSCLC tissues and cell lines compared with normal was associated with a poor survival rate in NSCLC patients. The research demonstrated that miR-379 could combine with the 3′UTR of *CHUK* and obviously down-regulate its expression in NSCLC cells. MiR-379 played an important role in suppressing cell invasion of NSCLC through regulating EMT. Furthermore, *CHUK* was an oncogene and the overexpression of miR-379 could inhibit *CHUK* and NF-кB pathway to prevent NSCLC [87]. The low expression of miR-874 was detected in NSCLC tissues while its expression was relatively higher in corresponding adjacent nontumor tissues. The results from functional assays exerted that miR-874 could inhibit cell proliferation and mobility, while decreasing miR-874 in A549 and H1299 cells showed an opposite effect. Luciferase assay displayed that miR-874 down-regulated the expression of *AQP3* through a direct binding to the 3 ‘UTR of *AQP3* mRNA. MiR-874 inhibited *AQP3* and the PI3K/AKT signaling pathway, resulting in the inhibition of EMT in A549 cells. Additionally, miR-874 had a suppressive effect on NSCLC growth in vivo [88]. Recently, a study found miR-940 could inhibit the proliferation of NSCLC by targeting SNAI1 and inhibiting TGF-β-induced EMT and invasion in vitro. Namely, this finding indicated that knocking down SNAI1, which is regulated by miRNA-940, could repress TGF-β-induced EMT and migration and invasion of NSCLC [89].

### 3.6. Pancreatic Cancer

Screening of transcription profiles revealed that 113 microRNAs and 1749 messenger RNAs expressed differentially in pancreatic cancer tissues. In particular, the expression of miR-203a-3p was markedly down-regulated in both pancreatic cancer tissues and cells, while the expression of *SLUG* was opposite. Luciferase reporter assay and RNA immunoprecipitation confirmed that miR-203a-3p could directly target *SLUG*. Further study displayed that miR-203a-3p mimic transfection could suppress migration, invasion, and EMT process through sponging *SLUG* [90]. A recent study showed that miR-338-5p was lowly expressed in pancreatic cancer tissues and the expression was negatively correlated with lymph node metastasis. Additionally, miR-338-5p overexpression could inhibit EMT and the metastasis process of pancreatic cancer cells. Furthermore, miR-338-5p could target *EGFR* and suppress *EGFR/ERK* pathways. These findings suggested miR-338-5p as a negative regulator of the EMT process in pancreatic cancer [91].

### 3.7. Papillary Thyroid Cancer

Papillary thyroid carcinoma is the most common type of thyroid cancer, with an increasing morbidity rate in recent years [92]. Mechanically, miR-451a played an important role in suppressing proliferation, EMT, and apoptosis in papillary thyroid cancer cells in vitro. Furthermore, luciferase reporter assays proved that miR-451a could bind to 3′UTR of *PSMB8* directly. In addition, in vitro, functional experiments showed miR-451a could suppress the ontogenetic capabilities of PTC cells by targeting *PSMB8* as well, which was consistent with that of in vivo [93]. Research revealed that the expression of miR-630 was decreased in papillary thyroid carcinoma cell lines TIPC-1 and SW1736 cells. Up-regulation of miR-630 could suppress the cell proliferation and promote cell apoptosis in PTC by inhibiting the expression of caspase-3 and caspase-6, combined with inhibiting PTC cell migration and invasion through preventing EMT progress. Moreover, enhanced miR-630 expression reduced protein phosphorylation levels in the *JAK2/STAT3* signaling pathway [94].

### 3.8. Other Types of Cancer

It has been observed that the expression level of miR-499a-5p was decreased in cervical cancer tissues and cell lines. The research showed increased expression of miR-499a-5p inhibited cervical cancer cell proliferation, invasion, migration, and EMT by targeting the gene *eIF4E* [95]. MiR-373-3p could inhibit EMT progress in choriocarcinoma JEG-3 and JAR cells through the inhibition of the TGF-β signaling pathway. MiR-373-3p impede the migratory and invasive ability of choriocarcinoma cells. Microarray analysis indicted an interaction between miR-373-3p and TGF-β R2, and luciferase assay validated that miR-373-3p could directly bind to TGF-β R2 [96]. MiR-4429 was found to express lowly in clear cell renal cell carcinoma tissues and cells and could inhibit cell proliferation, migration, and invasion combined with the EMT process. Further study indicated that miR-4429 targeted *CDK6* and had a negative correlation with it [97]. MiR-34a-5p played an anti-esophageal squamous cell carcinoma (ESCC) role by targeting the Hippo-YAP1/TAZ signaling pathway while the expression of LEF1 was high in ESCC tissue and cell lines. Inexplicably, high expression of miR-34a-5p promoted cancer cell viability, while up-regulation of *LEFA1* could impair the inhibitory effect of miR-34a-5p on ESCC cell lines [98]. Real-time PCR results showed that miR-22 was expressed at a low level in ovarian cancer tissues and cells, and the expression of miR-22 had a negative correlation with poor prognosis. *NLRP3* was up-regulated in ovarian cancer, and the expression of *NLRP3* was negatively correlated with miR-22. Further study showed that miR-22 inhibited cell viability and EMT through direct targeting of *NLRP3* mRNA, and the *NLRP3* overexpression partially reversed the effects of miR-22 [99]. *GRHL2* could maintain the epithelial phenotype epithelial status of ovarian cancer. ChIP-seq pointed that *GRHL2* could bind to the promoter site of miR-200b/a, and *ZEB1* was the direct target of miR-200b/a. Then, GRHL2 and ZEB1 formed a negative feedback to suppress EMT process through the regulation of miR-200b/a [100]. The expression of miR-490-5p in primary pharyngolaryngeal cancer tissues and cell lines including BICR 18 and FaDu cells was obviously decreased. Research demonstrated that the high levels of *MAP3K9* could facilitate cell viability, migration and invasion rates, the EMT process, and ability of cloning. Furthermore, miR-490-5p was found to target *MAP3K9* and then regulate cell proliferation, migration, invasion, and EMT in pharyngolaryngeal cancer [101]. IL-8 acted as a tongue squamous cell carcinoma (TSCC) growth factor through promoting invasion and EMT of TSCC cells, which could be suppressed by miR-940 up-regulation. Interestingly, if gene *CXCR2* was silenced or NF-кB inhibitor existed, the positive effect of miR-940 disappeared as well, suggesting that miR-940 regulated the growth of TSCC cells through targeting *CXCR2*/NF-кB system [102].

## 4. Compounds Regulated miRNAs as Potential Therapeutics for Inhibiting EMT

In recent years, development with regard to cancer research in the field of miRNAs has increased remarkably. Gradually, miRNA-based therapeutics have been attempted in the preclinical and clinical stages, while miRNA-based cancer diagnostics and prognostics have been approached clinically for cancer treatment management. Notably, numerous pure compounds exhibit multiple anti-cancer activities through the regulation of miRNAs, even in the EMT process [21,103]. Examples of pure compounds known to regulate miRNAs associated with EMT progression are presented below. 

### 4.1. Curcumin

Autocrine GH was an oncogene that activated the expression of miR-182-96-183 cluster and promoted EMT in breast cancer. It was shown that curcumin could inhibit cell growth and proliferation induced by autocrine GH in T47D cells. Further studies showed that long-term treatment with curcumin prevented invasion and metastasis induced by autocrine GH and EMT activation in T47D cells by inhibiting NF-кB signaling and expression of the miR-182-96-183 cluster. At the same time, curcumin induced apoptosis in T47D cells by regulating BCl-2 family members [104]. Curcumin could repress cell migration, invasion, and the EMT process through the up-regulation of miR-200c in colorectal cancer. Further study displayed that miR-200c could directly target EPM5, and decreased EPM5 could contribute to curcumin-induced inhibition on EMT progression. Moreover, high expression of EPM5 was positively correlated with advanced TNM stages and poor survival in CRC patients [105]. Curcumin had anti-tumor effects and could up-regulate the expression of miR-206 in CRC cells. Furthermore, miR-206 could directly bind to *SNAI2* and inhibit its expression so that it inhibits EMT in HCT116 cells. Meanwhile, curcumin suppressed EMT of CRC cells via up-regulation of miR-206 expression [106]. 

### 4.2. Propofol

Propofol critically suppressed cell proliferation, migration, invasion, and the EMT process in TPC-1 and IHH-4 papillary thyroid carcinoma cells. Interestingly, miR-122 was up-regulated after propofol treatment, and overexpressed miR-122 exerted a similar effect on papillary thyroid carcinoma cells. Transfecting miR-122 mimic could at least partly block the inhibitory effect of propofol [107]. Propofol could inhibit hypoxia-induced esophageal cancer cell migration, invasion, and EMT. Decreased miR-498 would at least partly reverse the suppressive effect of propofol on esophageal cancer progression. Luciferase reporter assay and RNA immunoprecipitation assay displayed a direct binding between lncRNA TMPO-AS1 and miR-498. High expression of TMPO-AS1 could also weaken the effect of propofol in esophageal cancer, suggesting that propofol might block esophageal cancer development through the regulation of TMPO-AS1/miR-498 axis [108].

### 4.3. Shikonin

Shikonin inhibited the migratory and invasive ability of MDA-MB-231 and BT549 cells, and the EMT process was suppressed in MDA-MB-231 cells after shikonin treatment. Mechanisms might be attributed to the inhibition of miR-17-5p by shikonin. Further study showed that miR-17-5p was overexpressed in breast cancer. Luciferase reporter assays exerted that miR-17-5p could directly bind to *PTEN* [109]. Shikonin showed anti-tumor activity in hepatocellular carcinoma cell lines Huh7 and HepG2 and could induce miR-106a reduction in later treatment. Inexplicably, down-regulation of miR-106b could enhance the inhibitory effect of shikonin on hepatocellular carcinoma cell migration and EMT. Further study suggested that the expression of SMAD7 by shikonin led to TGF-β signaling pathway activation, resulting in the decrease in miR-106a. This finding indicated that inhibiting miR-10b in shikonin-treated hepatocellular carcinoma would increase shikonin’s anti-cancer effects. However, the mechanism of shikonin-induced miR-106a and the synergistic effect of miR-106a inhibitor and shikonin in vivo should be further analyzed [110].

### 4.4. Other Drug Candidates

Apigenin inhibited proliferation, invasion, and EMT cervical carcinoma Hela and CaSki cells. Low expression of miR-152-5p was detected in cervical carcinoma tissue samples from patients, while the expression of *BRD4* was relatively low. Interestingly, transfection of the miR-152-5p inhibitor could at least partly reverse the regulatory effect of apigenin on cervical carcinoma cells, while overexpressed *BRD4* exerted a similar effect. These findings suggest that apigenin suppresses cervical carcinoma development through miR-152-5p/BRD4 axis [111]. Gastric cancer cell migration, invasion, and EMT were reduced by crocin in vitro. Mechanically, anti-oncogenic miR-320 was elevated by crocin, coupled with a decrease in the transcription factor *KLF5* which is the targeted gene of miR-320 and HIF-1α. Increasing *KLF5* expression reversed the inhibition effects of crocin in gastric cancer cells. Overall, the results confirmed that crocin affects GC through miR-320/KLF5/HIF-1α signaling [112]. Ginsenoside Rg3 showed anti-tumor activity in human oral squamous cell carcinoma SCC-9 and HSC-5 cells and could down-regulate the expression of miR-221 in later treatment. Research suggested that Ginsenoside Rg3 inhibited cell viability, proliferation, and the EMT process while it promoted cell apoptosis through inactivating PI3K/AKT and MAPK/ERK pathways in SCC-9 cells. *TIMP3* was a target gene of miR-221 and could modulate PI3K/AKT and MAPK/ERK pathways. Thus, decreasing miR-221 expression could enhance the anticancer effect of Ginsenoside Rg3. Moreover, the volume of oral squamous cell carcinoma orthotopic murine model was reduced by Ginsenoside Rg3 in an in vivo experiment [113]. Isoliquiritigenin (ISL) could inhibit TNBC cell proliferation, migration, and invasion. PCR array screened an essential increase in miR-200c in BT-549 and MDA-MB-231 cells after ISL interference. Further study showed that ISL could inhibit the EMT process in TNBC through up-regulating miR-200c, and ISL exerted an inhibitory effect on metastasis and tumor growth in nude mice models by increasing miR-200c. C-JUN was identified as the direct target of miR-200c according to the results of luciferase assay. Additionally, ISL might promote miR-200c expression by demethylating the miR-200c promoter region [114]. Interestingly, metformin could enhance the chemosensitivity of pancreatic cancer cells to gemcitabine and inhibit the EMT process induced by gemcitabine. A mechanism study revealed that metformin promoted the chemosensitivity and suppressed EMT in pancreatic cancer through the up-regulation of miR-663. MiR-663 was hypomethylated and after treatment of metformin, the expression of miR-663 was increased. Furthermore, overexpressed miR-663 reversed EMT and promoted chemosensitivity by directly targeting TGF-β1 [115]. Interestingly, EMT displays a regulatory effect of tumor chemoresistance in nasopharyngeal carcinoma. Taxol-resistant cell lines 5-8F/Taxol and CNE-1/Taxol had the high capacity to go through the EMT process. Microarray analysis exerted a significant decrease in 5-8F/Taxol cells after neferine (NEF) treatment. Further study revealed that NEF is found to sensitize cancer cells to Taxol in nasopharyngeal carcinoma through down-regulation of miR-130b-5p. Moreover, high expression of miR-130b-5p would block the inhibitory effect of NEF on EMT-associated metastatic ability and increase the chemotherapy resistance to Taxol [116]. This finding indicated that puerarin could upregulate *PTEN* expression by exerting a negative influence on miR-21, resulting in the inhibition of the EMT process in hepatocellular carcinoma cell lines. In addition, tumor sizes were smaller and there were fewer nodules in the livers of nude mice in the puerarin group [117]. Sevoflurane (SEV) could inhibit breast cancer migration, invasion, and the EMT process. A detailed mechanism study revealed that SEV could up-regulate miR-139-5p and down-regulate *ARF6*, and the decrease in miR-139-5p and increase in *ARF6* could at least partly reverse the suppressive effect of SEV on breast cancer. Further study exerted that miR-139-5p was expressed at a low level in breast cancer tissues compared with corresponding normal tissues, while *ARF6* was expressed at a relatively high level. Meanwhile, luciferase assay validated that *ARF6* was the direct target of miR-139-5p [118]. 

## 5. Conclusions and Perspective

MiRNAs in cancer, exert as potential targets of efficient target therapy and effective pro-diagnosis for cancer patients. In cancer cells, EMT functions as the initial measure of migration and invasion in most cases, while undergoing MET facilitates metastatic colonization during secondary tumor formation [119]. It is interesting to find that EMT is also associated with chemoresistance in many different preclinical models with unclear mechanisms, until the relationship of EMT and cancer stem cells is investigated [120,121]. Recent studies point out that cancer cells undergoing the EMT process display stem-like features [122,123]. Furthermore, EMT-TFs, belonging to SNAIL, TWIST, and ZEB families, are responsible for the promotion of cancer stemness in various types of cancer, including nasopharyngeal carcinoma, breast cancer, colon cancer, etc. [124,125,126,127]. Additionally, the Wnt signaling pathway, Notch/Jagged pathway, and hedgehog signal pathway play important roles both in EMT and cancer stem cells [128,129,130]. Thus, researchers in increasing number make an effort to explore the roles of varied miRNAs associated with EMT in cancer [131].

Accumulating evidence points out that some miRNAs could act as oncogenes, while some display an inhibitory effect on the EMT process in cancer. Interestingly, some trmiRs could directly bind to 3′UTR of EMT-TFs, inhibiting EMT development. However, the effects of oncomiRs on EMT suppressors, such as OVOL1/2, are still undetermined. Notably, recent studies exerted that some oncomiRs could promote EMT, leading to drug resistance via the promotion of stemness, while some trmiRs could inhibit EMT, sensitizing cancer cells to chemotherapy mainly through the inhibition of EMT-TFs [120,122]. With the development of chemotherapy from natural compounds, it is worthwhile to screen pure compounds from natural plants as the drug candidate for cancer therapy by down-regulating oncomiRs, up-regulating tsmiRs, and modulating deregulated miRNAs in the tumor microenvironment [21,132,133]. It is also of interest to investigate whether the role of miRNAs, namely oncogenic or tumor suppressive, is analogous even in different types of cancer. Further studies investigating the synergetic and antagonistic effects of miRNAs and compounds are essential for complete evaluation in order to provide a more comprehensive understanding of emerging challenges prior to clinical application.

## Figures and Tables

**Figure 1 ijms-22-07526-f001:**
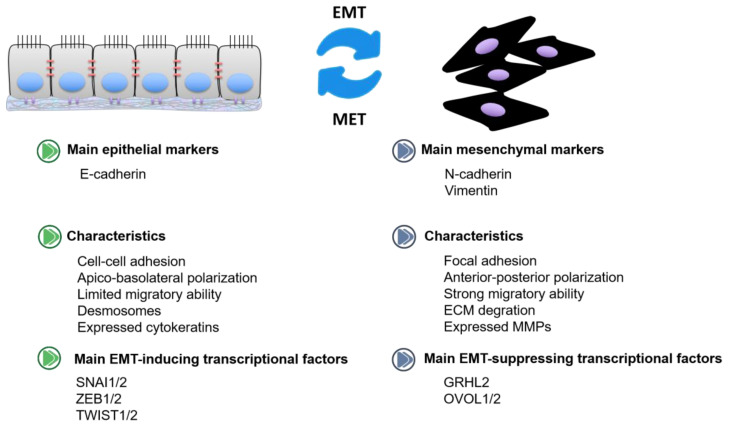
The characteristics of epithelial-like cells and mesenchymal-like cells in cancer.

**Figure 2 ijms-22-07526-f002:**
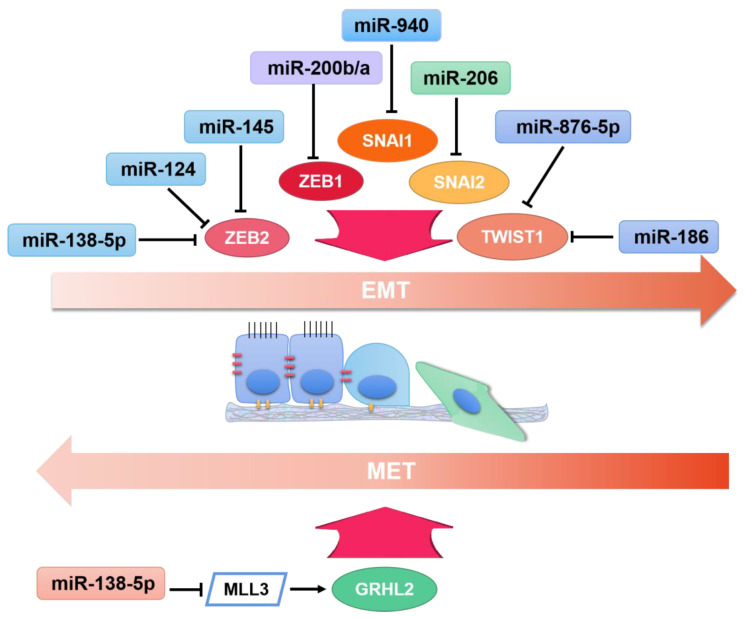
The regulation of microRNAs on transcription regulators impacting EMT, namely EMT-inducers and EMT-suppressors in cancer.

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
