# Peer review of "MicroRNAs in Epithelial–Mesenchymal Transition Process of Cancer: Potential Targets for Chemotherapy"

_ijms, 2021, doi:10.3390/ijms22147526_

Round 1
Reviewer 1 Report
Concise Summary
EMT transition is known to play an important role in cancer progression and drug resistance. The authors review the role of microRNAs in the epithelial-mesenchymal transition (EMT) process of cancer, either promoting or inhibiting EMT progression. In addition, the authors discuss about the effect of some drugs obtained from natural plants on EMT, which could result in the inhibition of cancer development.
Final consideration
It is a complete review of state of art of microRNAS on EMT in different kinds of cancer. This is a exhaustive and well-written text, and it provides an up to date about possible mechanisms of interference of microRNAs in the EMT phenomenon. The molecular targets of miRNAs and their underlying signaling pathways are also explored comprehensively.
The text is ordered but its content is not especially relevant. The effect of product compounds on microRNAs is interesting but it is not presented with an original approach. It had been desirable to complete the explanation of the issue with original schemes that would have helped the reader to a better understanding of the action mechanism of microRNAs in cancer. Otherwise, it could result in a heavy reading for the reader who do not work this topic. The part of the effect of drugs compounds of natural plants is a minor part of the article and it does not give relevant considerations. Finally, the conclusions of the article are not informative enough.
Author Response
Response to Reviewer 1 Comments
Point 1:It is a complete review of state of art of microRNAS on EMT in different kinds of cancer. This is a exhaustive and well-written text, and it provides an up to date about possible mechanisms of interference of microRNAs in the EMT phenomenon. The molecular targets of miRNAs and their underlying signaling pathways are also explored comprehensively.
A: Sincere thanks to the reviewer. We are very grateful for the important comments from the reviewer to help us to improve our work.
Point 2:The text is ordered but its content is not especially relevant. The effect of product compounds on microRNAs is interesting but it is not presented with an original approach. It had been desirable to complete the explanation of the issue with original schemes that would have helped the reader to a better understanding of the action mechanism of microRNAs in cancer. Otherwise, it could result in a heavy reading for the reader who do not work this topic. The part of the effect of drugs compounds of natural plants is a minor part of the article and it does not give relevant considerations. Finally, the conclusions of the article are not informative enough.
A: Thank the reviewer ever so much. The corresponding content has been revised according to the essential advice. The action mechanism of microRNAs has been enlarged. The correlation between miRNA and potential drug candidates has been extended. We are making efforts to display the significance of microRNAs in EMT process as targets and the effect of compounds on EMT-associated microRNAs as drug candidates. More information and indication have been added into the conclusion part.
Reviewer 2 Report
This is a well-written reviews of miRNAs in cancer EMT field. Some concerns are listed below, which hopefully to be addressed;
- During the process of cancer metastasis, cancer cells are thought to need to proceed colony formation, which requires MET (mesenchymal epithelial transition) process. On this point, please provide some (experimental and clinical) evidences on how the EMT-inhibiting miRNAs work.
- Besides the EMT- (inducing) transcriptional factors (EMT-TFs), some transcriptional factors have been identified to cause MET, such as GRHL2 and Ovol2. Among these EMT and MET drivers, please provide a compact summary on how the miRNAs are working associated with these factors.
- EMT process of cancer cells is associated with cancer stemness and chemoresistance, which might be the most critical points on why EMT matters in cancer biology. Please provide a compact summary on how the miRNAs can modulate these points.
Author Response
Response to Reviewer 2 Comments
Point 1:During the process of cancer metastasis, cancer cells are thought to need to proceed colony formation, which requires MET (mesenchymal epithelial transition) process. On this point, please provide some (experimental and clinical) evidences on how the EMT-inhibiting miRNAs work.
A: Great thanks to the reviewer. The corresponding content has been revised according to the important suggestions. The background about MET has been added into introduction part. The regulation of EMT-inhibiting miRNAs has been added in tsmiRs part.
Point 2:Besides the EMT- (inducing) transcriptional factors (EMT-TFs), some transcriptional factors have been identified to cause MET, such as GRHL2 and Ovol2. Among these EMT and MET drivers, please provide a compact summary on how the miRNAs are working associated with these factors.
A: Thank the reviewer very much. The corresponding content has been revised according to the essential comment. The detail information about EMT-TF and EMT suppressors has been added in the introduction. The regulation of microRNAs on GRHL2 has been added. The regulatory network has been summarized as Figure 2. However, according to the literature review, the effect of EMT-associated microRNAs on Ovol2 is lacking.
Point 3: EMT process of cancer cells is associated with cancer stemness and chemoresistance, which might be the most critical points on why EMT matters in cancer biology. Please provide a compact summary on how the miRNAs can modulate these points.
A: We are very thankful for the great suggestions from the reviewer. The corresponding content has been revised according to the significant advice. The detail information about EMT, cancer stemness and chemoresistance has been added in the conclusion. The regulation of microRNAs on these points has been discussed and described.
Round 2
Reviewer 1 Report
I estimate that the review article has now more quality. The authors have made an effort to adapt the text to the reviewer's suggestions. The authors have incorporated into the text interesting information and new illustrative figures of the epithelial-mesenchymal transformation process. The article as a whole does not provide a very original view of the topic, but it is a clear update of current knowledge on the subject.